# Reduced metabolism supports hypoxic flight in the high-flying bar-headed goose (*Anser indicus*)

Jessica U Meir[1,2]*, Julia M York[2,3]*, Bev A Chua[2], Wilhelmina Jardine[2], Lucy A Hawkes[4], William K Milsom[2]

[1]NASA Johnson Space Center, Houston, United States; [2]University of British Columbia, Vancouver, Canada; [3]University of Texas at Austin, Austin, United States; [4]Hatherly Laboratories, College of Life and Environmental Sciences, University of Exeter, Exeter, United Kingdom

**Abstract** The bar-headed goose is famed for migratory flight at extreme altitude. To better understand the physiology underlying this remarkable behavior, we imprinted and trained geese, collecting the first cardiorespiratory measurements of bar-headed geese flying at simulated altitude in a wind tunnel. Metabolic rate during flight increased 16-fold from rest, supported by an increase in the estimated amount of $O_2$ transported per heartbeat and a modest increase in heart rate. The geese appear to have ample cardiac reserves, as heart rate during hypoxic flights was not higher than in normoxic flights. We conclude that flight in hypoxia is largely achieved *via* the reduction in metabolic rate compared to normoxia. Arterial $P_{O_2}$ was maintained throughout flights. Mixed venous $P_{O_2}$ decreased during the initial portion of flights in hypoxia, indicative of increased tissue $O_2$ extraction. We also discovered that mixed venous temperature decreased during flight, which may significantly increase oxygen loading to hemoglobin.
DOI: https://doi.org/10.7554/eLife.44986.001

*For correspondence:
jessica.u.meir@nasa.gov (JUM);
juliayork@utexas.edu (JMY)

**Competing interests:** The authors declare that no competing interests exist.

## Introduction

Flapping flight in birds is the most metabolically costly form of locomotion in vertebrates (*Butler and Bishop, 2000*). These costs are exacerbated with increasing elevation as the air becomes less dense, reducing oxygen available to support metabolism and requiring changes to the wing kinematics of forward flying birds (*Feinsinger et al., 1979*; *Dudley and Chai, 1996*; *Ellington, 1984*; *Pennycuick, 2008*). Our understanding of the adaptations that support high-altitude flight in birds is growing, particularly in the bar-headed goose, *Anser indicus* (*Storz et al., 2010*; *Scott et al., 2015*). This species migrates biannually across the Himalayan Mountains and Tibetan Plateau, wintering in India and breeding in China and Mongolia, typically flying through passes 5,000 to 6,000 m above sea level, where partial pressures of oxygen are only half of those at sea level. They have been documented flying at altitudes as high as 7,290 m (*Bishop et al., 2015*; *Hawkes et al., 2013*).

The physiological adaptations to hypoxia that have been previously described in bar-headed geese have been examined in birds at rest or running (*Fedde et al., 1989*; *Scott et al., 2015*; *Hawkes et al., 2014*; *Scott and Milsom, 2007*). Such adaptations are distributed throughout the oxygen transport cascade, the steps involved in oxygen transfer from atmosphere to mitochondria (ventilation, lung oxygen diffusion, circulation and tissue oxygen extraction) (*Scott et al., 2015*). Direct and integrated physiological measures of oxygen transport during flight, on the other hand, are extremely limited (*Butler and Bishop, 2000*; *Ward et al., 2002*), and none have been made under hypoxic conditions.

**Table 1.** Compiled flight, respiratory, and cardiovascular data from all three $O_2$ levels tested during rest (sitting bird, often asleep), pre-flight (standing bird just before flight), and steady-state flight (steady-state determined by $CO_2$ production).

Values are mean ± SEM. Asterisks indicate significant difference from normoxia (linear mixed model ANOVA; * indicates $p < 0.05$; ** indicates $p < 0.01$; *** indicates $p < 0.001$).

| | Normoxia 0.21 $F_iO_2$ | Moderate hypoxia 0.105 $F_iO_2$ | Severe hypoxia 0.07 $F_iO_2$ |
|---|---|---|---|
| Flight length (sec) Mean ± se (range) | 195 ± 12 (55−663) | 174 ± 14 (54−826) | 139 ± 29** (60−468) |
| n Birds | 7 | 6 | 3 |
| n Flights | 113 | 74 | 13 |
| Wingbeat frequency (Hz) | 4.97 ± 0.27 | 4.91 ± 0.28 | (-) |
| **Rest:** | | | |
| $\dot{V}_{O2}$ (ml $O_2$ min$^{-1}$ kg$^{-1}$) | 12.5 ± 0.4 | (-) | (-) |
| $\dot{V}_{CO2}$ (ml $CO_2$ min$^{-1}$ kg$^{-1}$) | 10.1 ± 0.5 | 7.9 ± 0.5 | 9.1 ± 0.7 |
| RER | 0.80 ± 0.02 | (-) | (-) |
| Heart rate (bpm) | 126.4 ± 7.4 | 108.1 ± 4.1 | 149.2 ± 24** |
| $CO_2$ pulse (ml $CO_2$ beat$^{-1}$ kg$^{-1}$) | 0.085 ± 0.003 | 0.074 ± 0.003 | 0.068 ± 0.005 |
| **Pre-flight:** | | | |
| $\dot{V}_{O2}$ (ml $O_2$ min$^{-1}$ kg$^{-1}$) | 16.0 ± 0.6 | (-) | (-) |
| $\dot{V}_{CO2}$ (ml $CO_2$ min$^{-1}$ kg$^{-1}$) | 13.6 ± 0.6 | 10.0 ± 0.7 | 9.69 ± 1.9 |
| RER | 0.86 ± 0.02 | (-) | (-) |
| Heart rate (bpm) | 114.2 ± 2.9 | 120.8 ± 3.5 | 117.0 ± 7.9 |
| $CO_2$ pulse (ml $CO_2$ beat$^{-1}$ kg$^{-1}$) | 0.11 ± 0.004 | 0.081 ± 0.006 | 0.071 ± 0.016 |
| **Flight:** | | | |
| $\dot{V}_{O2}$ (ml $O_2$ min$^{-1}$ kg$^{-1}$) | 222.6 ± 3.5 | (-) | (-) |
| $\dot{V}_{CO2}$ (ml $CO_2$ min$^{-1}$ kg$^{-1}$) | 222.2 ± 4.9 | 186.8 ± 4.2*** | 126.1 ± 4.3*** |
| RER | 0.99 ± 0.01 | (-) | (-) |
| Heart rate (bpm) | 313.3 ± 4.1 | 312.1 ± 6.7 | 329.0 ± 14.3 |
| $CO_2$ pulse (ml $CO_2$ beat$^{-1}$ kg$^{-1}$) | 0.72 ± 0.02 | 0.61 ± 0.02*** | 0.45 ± 0.04*** |

DOI: https://doi.org/10.7554/eLife.44986.003

*Ward et al. (2002)* documented that the metabolic cost of flight in bar-headed geese in normoxia at sea-level in a wind tunnel was roughly 12 times resting metabolic rate. This was associated with an approximately two-fold increase in heart rate. Based on extrapolation from wind tunnel heart rate data, flight metabolic rate for birds migrating at an altitude around 6,000 m in the wild was calculated to be approximately 15 times resting metabolic rate (*Bishop et al., 2015*). This further increase in metabolic cost is concordant with the increased biomechanical costs of flying in the thinner air at high altitude (requiring increased flight speeds to offset reductions in lift; *Pennycuick, 2008*) but may also arise in part from increased metabolic demands on the cardiorespiratory system associated with flight in hypoxia. Bar-headed geese trained to run on a treadmill did not show a significant change in metabolic rate between normoxia and severe hypoxia, however the increase in metabolic rate from rest to running was only ~2.5 fold (*Hawkes et al., 2014*). Whether or not hypoxia increases the metabolic cost of flight remains to be determined.

Based on these observations, we aimed to determine (1) how the metabolic challenge of flight differs between normoxia and normobaric hypoxia, and (2) whether bar-headed geese are capable of wind tunnel flight in severe normobaric hypoxia equivalent to altitudes of roughly 9,000 m

(0.07 $F_iO_2$), the maximum altitude at which they have been anecdotally reported to fly (*Swan, 1961*).

## Results

Descriptive statistics are reported in *Table 1* and supplementary files, mean ± SEM is reported here unless estimated marginal mean (*EMM* ± SEM) or median is indicated. Without instrumentation, birds flew for up to 45 min in the wind tunnel, however, once fully instrumented, experimental flights were much shorter. There was a significant effect of oxygen level on flight duration ($F_{2, 363.35}$=6.55, p=0.0016). In the post-hoc comparison, severe hypoxia (0.07 $F_iO_2$, equivalent to ~ 9,000 m) was significantly shorter with an estimated marginal mean (*EMM*) of 79.1 ± 36.6 s compared to an *EMM* of 187.7 ± 20.7 s in normoxia (t = −3.245, p=0.0039). Moderate hypoxia (0.105 $F_iO_2$, equivalent to ~ 5,500 m) had an *EMM* of 148.9 ± 22.3 s, and was not significantly different from normoxia (t = −2.315, p=0.0635). The intraclass correlation coefficient (ICC) for this model was 0.141. Only one bird (bird 45) flew in severe hypoxia consistently, with a median duration of 100 s. This was significantly shorter (one-way ANOVA on ranks; $H_{(2)}$=14.911, p<0.001; post-hoc Dunn's method Q = 3.815, p<0.05) than this bird would fly in normoxia (median = 232.5 s) but not moderate hypoxia (median = 158 s, Q = 2.113, p>0.05; *Supplementary file 1*).

That only one bird consistently flew in severe hypoxia likely results in a survivor bias of the severe hypoxia data, as that bird may have flown more consistently due to a greater ability to cope with the metabolic challenge. We attempt to correct this bias by allowing for comparison across all oxygen levels in this one bird (*Supplementary file 1*) as well as plotting data for individual birds (*Figure 1—figure supplement 1*). However, this should be taken into consideration especially in comparisons of moderate versus severe hypoxia.

Wing-beat frequency was measured in a separate biomechanical study and was similar regardless of oxygen level (mean 4.97 ± 0.27 Hz in normoxia and 4.91 ± 0.28 Hz in moderate hypoxia, *Supplementary file 4*; *Whale, 2012*).

### Metabolic rate

The respiratory exchange ratio RER ($\dot{V}_{CO_2}/ \dot{V}_{O_2}$ = RER) could only be measured in normoxia due to unreliable $\dot{V}_{O_2}$ values in hypoxia. There was a significant effect of activity on RER ($F_{2, 301.95}$=54.37, p<0.0001, ICC=0.254). RER in flight (*EMM* of 1.00 ± 0.034) was significantly higher than pre-flight (*EMM* of 0.87 ± 0.035; t=7.026, p<0.0001) and rest (*EMM* of 0.80 ± 0.035; t=10.073, p<0.0001). RER in pre-flight was also significantly higher than at rest (t=3.453, p=0.0019).

$\dot{V}_{CO_2}$ differed significantly based on oxygen level in flight ($F_{2, 549.54}$=74.155, p<0.0001, ICC=0.145), but not at rest or during pre-flight (p>0.468). Within flight data, $\dot{V}_{CO_2}$ in normoxia (*EMM*=223.8 ± 4.8 mL $CO_2$ $min^{-1}$ $kg^{-1}$) was significantly higher (t=−8.047, p<0.0001) than $\dot{V}_{CO_2}$ in moderate hypoxia (*EMM*=193.0 ± 5.1 mL $CO_2$ $min^{-1}$ $kg^{-1}$). $\dot{V}_{CO_2}$ dropped significantly in severe hypoxia (*EMM*=142.5 ± 8.3 mL $CO_2$ $min^{-1}$ $kg^{-1}$) compared to moderate hypoxia (t=−6.562, p<0.0001). Individual minimum metabolic rate (the lowest steady state $\dot{V}_{CO_2}$ of all flights for each bird) was not different in normoxia and moderate hypoxia (paired t-test; t=0.157; p=0.883).

Heart rate did not differ significantly between $O_2$ levels in flight ($F_{2, 441.13}$=1.237, p=0.2914, ICC = 0.166), but $O_2$ level had a marginally significant effect on heart rate pre-flight ($F_{2, 441.53}$=3.077, p=0.0471). However, during post-hoc testing, no comparisons were significant within pre-flight (p>0.12). There was a significant effect of $O_2$ level on heart rate at rest ($F_{2, 439.20}$=7.688, p=0.0005), because the resting heart rate in severe hypoxia (*EMM* = 149.7 ± 11.9 beats $min^{-1}$) was significantly higher (t = −2.569, p=0.0316) than in normoxia (*EMM* = 128.3 ± 9.2 beats $min^{-1}$) and moderate hypoxia (t = 3.817, p=0.0005). Resting heart rate in moderate hypoxia (*EMM* = 107.3 ± 10.1) did not differ significantly from normoxia (t = 2.077, p=0.1151).

There was a significant effect of oxygen level on $CO_2$ pulse for flight data ($F_{2, 450.00}$ = 31.845, p<0.0001, ICC = 0.162) but not on pre-flight or rest data (p>0.61). $CO_2$ pulse in normoxic flight (*EMM* = 0.722 ± 0.021 mL $CO_2$ $beat^{-1}$ $kg^{-1}$) was significantly higher (t = −5.818, p<0.0001) than $CO_2$ pulse in moderate hypoxic flight (*EMM* = 0.627 ± 0.022 mL $CO_2$ $beat^{-1}$ $kg^{-1}$). $CO_2$ pulse in

moderate hypoxic flight was significantly higher (t = −3.666, p=0.0008) than in severe hypoxia ($EMM = 0.514 \pm 0.034$ mL $CO_2$ $beat^{-1}$ $kg^{-1}$).

## Correlations

Due to the non-independence of our repeated measurements across individual birds, we cannot calculate correlation statistics such as $r^2$. In comparing the correlation of heart rate versus metabolic rate we generated a linear mixed model for the combined data and found heart rate was a significant predictor of metabolic rate (df = 444.7, t = 37.535, p<0.0001, ICC = 0.143). However, when we added activity as a fixed effect, heart rate was no longer a significant predictor of metabolic rate in flight (df = 442.9, t = 0.244, p=0.808, ICC = 0.127), only during pre-flight (df = 446.2, t = −5.113, p<0.0001, ICC = 0.106) and rest (df = 444.9, t = 18.652, p<0.0001, ICC = 0.184). This indicates that, when pooled, the data are bimodal (flight and preflight/rest), but within the flight data, there is a large variation in $CO_2$ production at any level of heart rate and vice versa (*Figure 1*, and by individual in *Figure 1—figure supplement 1* and *Figure 1—figure supplement 2*).

Two other studies (*Ward et al., 2002*; *Hawkes et al., 2014*) have measured metabolic rates and heart rates in resting and exercising bar-headed geese. We pooled our complete data set with those values (*Figure 1B*) and also compared the distribution of our heart rate data to those measured in migrating wild geese (*Bishop et al., 2015*; *Figure 2*). The comparison shows a remarkable agreement in the peaks of the heart rate measurement distribution of geese flying below 2,300 meters in the wild (*Bishop et al., 2015*) and in the present study (but note possible survivor bias for severe hypoxia data).

## Blood gases

We measured mixed venous $P_{O_2}$ in normoxia, moderate hypoxia, and severe hypoxia and report values for several time points across the flight (pre-flight, start, steady state, end, and in recovery). There was a significant effect of both oxygen level ($F_{2, 79.197}$=22.3439, p<0.0001) and timepoint ($F_{4, 79.113}$=5.0645, p=0.0011) on venous $P_{O_2}$, but not the interaction $O_2$ level*timepoint ($F_{8, 79.127}$=0.9865, p=0.453). Venous $P_{O_2}$ did not significantly differ between exposed oxygen levels during pre-flight (preflight normoxia $EMM = 47.68 \pm 2.52$ mmHg, preflight moderate hypoxia $EMM = 44.47 \pm 3.16$ mmHG, preflight severe hypoxia $EMM = 38.68 \pm 4.21$ mmHg), but then was maintained in normoxia (start $EMM = 50.00 \pm 2.52$ mmHg) while dropping in both levels of hypoxia such that both moderate hypoxia (start $EMM = 34.71 \pm 3.16$ mmHg, t = −4.360, p=0.0001) and severe hypoxia (start $EMM = 33.61 \pm 4.21$; t = −3.705, p=0.0012) were significantly different from normoxia at the start of flight, but did not differ from each other (t = −0.236, p=1.0). During the steady state portion of the flight, $P_{O_2}$ in normoxia (steady state $EMM = 42.30 \pm 2.49$ mmHg) dropped slightly so moderate hypoxia (steady state $EMM = 33.59 \pm 3.40$ mmHg) was marginally non-significant (t = −2.373, p=0.0600) while $P_{O_2}$ in severe hypoxia (steady state $EMM = 29.61 \pm 4.21$) remained significantly different from normoxia (t = −2.881, p=0.0152). That pattern held through the end of the flight (end normoxia $EMM = 41.48 \pm 2.40$ mmHg, end moderate hypoxia $EMM = 33.54 \pm 3.40$, end severe hypoxia $EMM = 29.25 \pm 4.21$ mmHg), but in recovery $P_{O_2}$ in normoxia (recovery $EMM = 50.60 \pm 2.45$ mmHg) increased more than $P_{O_2}$ in hypoxia so that both moderate hypoxia (recovery $EMM = 38.13 \pm 3.16$ mmHg; t = −3.588, p=0.0017) and severe hypoxia (recovery $EMM = 34.85 \pm 4.21$ mmHg; t = −3.581, p=0.0017) were significantly lower (*Figure 3*, *Supplementary file 2*).

There was no significant effect of oxygen level on venous blood temperature ($F_{2, 72.225}$=1.7253, p=0.1854; *Figure 3B*) nor the interaction of $O_2$ level*timepoint ($F_{8, 71.045}$=0.3347, p=0.9497). There was a significant main effect of timepoint ($F_{4, 71.036}$=11.4269, p<0.0001), which held within each oxygen level (normoxia: $F_{4, 71.17}$=6.333, p=0.0002; moderate hypoxia: $F_{4, 71.01}$=3.547, p=0.0107; severe hypoxia: $F_{4, 71.01}$=3.497, p=0.0115). The effect of timepoint was due to a drop in venous temperature between preflight and the steady state portion of the flight: the minimum drop was 1.22 ˚C and the maximum drop was 2.72 ˚C. In normoxia, venous temperature was significantly higher in preflight (preflight $EMM = 41.37 \pm 0.402$˚C) compared to steady state (steady state $EMM = 39.99 \pm 0.441$˚C; t = 4.342, p=0.0005) and the end of the flight (end $EMM = 40.38 \pm 0.421$˚C; t=−3.404, p=0.0110), as well as in steady state compared to recovery (recovery $EMM = 41.04 \pm 0.403$˚C; t = 3.311, p=0.0146) and the start of the flight (start $EMM = 41.10 \pm 0.406$˚C; t=−3.443, p=0.0097). In

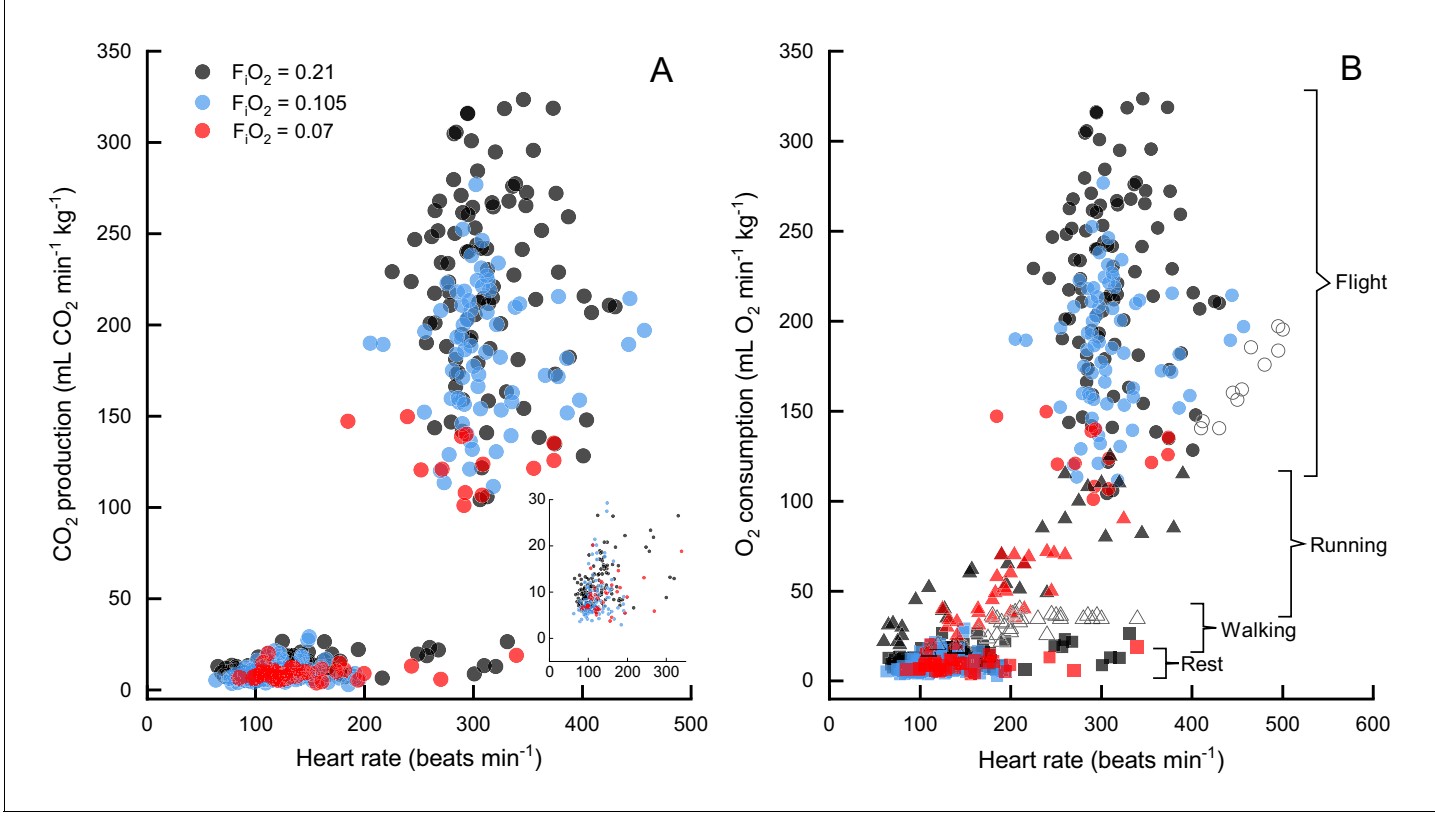

**Figure 1.** $CO_2$ production versus heart rate in $F_iO_2$=0.21 (black), $F_iO_2$=0.105 (blue), and $F_iO_2$=0.07 (red). Normoxia and moderate hypoxia data from this study shown in (**A**), inset shows expansion of data at rest. N=7 birds for all data, n=89 sessions for rest in normoxia, n=113 flights in normoxia, n=54 sessions for rest in moderate hypoxia, n=74 flights in moderate hypoxia, n=13 flights in severe hypoxia (note that only one bird flew consistently in severe hypoxia), n=29 sessions for rest in severe hypoxia. In (**B**) oxygen consumption versus heart rate for bar-headed geese from three studies, *Hawkes et al. (2011)* (running, filled triangles), *Ward et al. (2002)* (open circles are flight and open triangles are walking), and the present study (filled circles are flight data, filled squares are rest). Note that $\dot{V}_{O2}$ values for the current study have been calculated from $\dot{V}_{CO2}$ values, assuming an RER=1.
DOI: https://doi.org/10.7554/eLife.44986.004

The following figure supplements are available for figure 1:

**Figure supplement 1.** $CO_2$ production versus heart rate for rest and preflight (squares), as well as steady-state flight (circles) separated by individual birds in $F_iO_2$ = 0.21 (black), $F_iO_2$ = 0.105 (blue), and $F_iO_2$ = 0.07 (red).
DOI: https://doi.org/10.7554/eLife.44986.005

**Figure supplement 2.** Means and standard errors in $CO_2$ production and heart rate are plotted for flight data of individual birds, indicated by bird number.
DOI: https://doi.org/10.7554/eLife.44986.006

moderate hypoxia, venous temperature in steady state (steady state *EMM* = 39.998 ± 0.509°C) was significantly lower than both preflight (preflight *EMM* = 41.473 ± 0.509°C; t = 3.139, p=0.0247) and the start of the flight (start *EMM* = 41.373 ± 0.509°C; t=−2.926, p=0.0460). In severe hypoxia, however, no pairwise comparisons among timepoints were significant (p>0.07).

We successfully measured arterial (carotid) and mixed venous $P_{O2}$ at all three levels of oxygen for one bird (bird 45): normoxia (venous: four flights, arterial: two flights), moderate hypoxia (venous: four flights, arterial: three flights), and severe hypoxia (venous: three flights, arterial: four flights, *Supplementary file 1*). During preflight, there was a significant effect of oxygen exposure level on arterial $P_{O2}$ ($H_{(2)}$=6.0, p=0.014) from a mean of 72.1 ± 0.42 mmHg (median 72.1 mmHg) in normoxia, to a mean of 56.5 ± 5.4 mmHg (median 56.5 mmHg) in moderate hypoxia, and a mean of 36.7 ± 0.54 mmHg (median 36.5 mmHg) in severe hypoxia (*Supplementary file 3*). In post-hoc testing, no comparisons among preflight arterial $P_{O2}$ were significant (p>0.05). In general, preflight arterial $P_{O2}$ levels were maintained throughout flights. There was a significant effect of oxygen exposure level on arterial $P_{O2}$ during steady state flight ($F_{2,\,6}$=23.1294, p=0.002), such that arterial $P_{O2}$ in

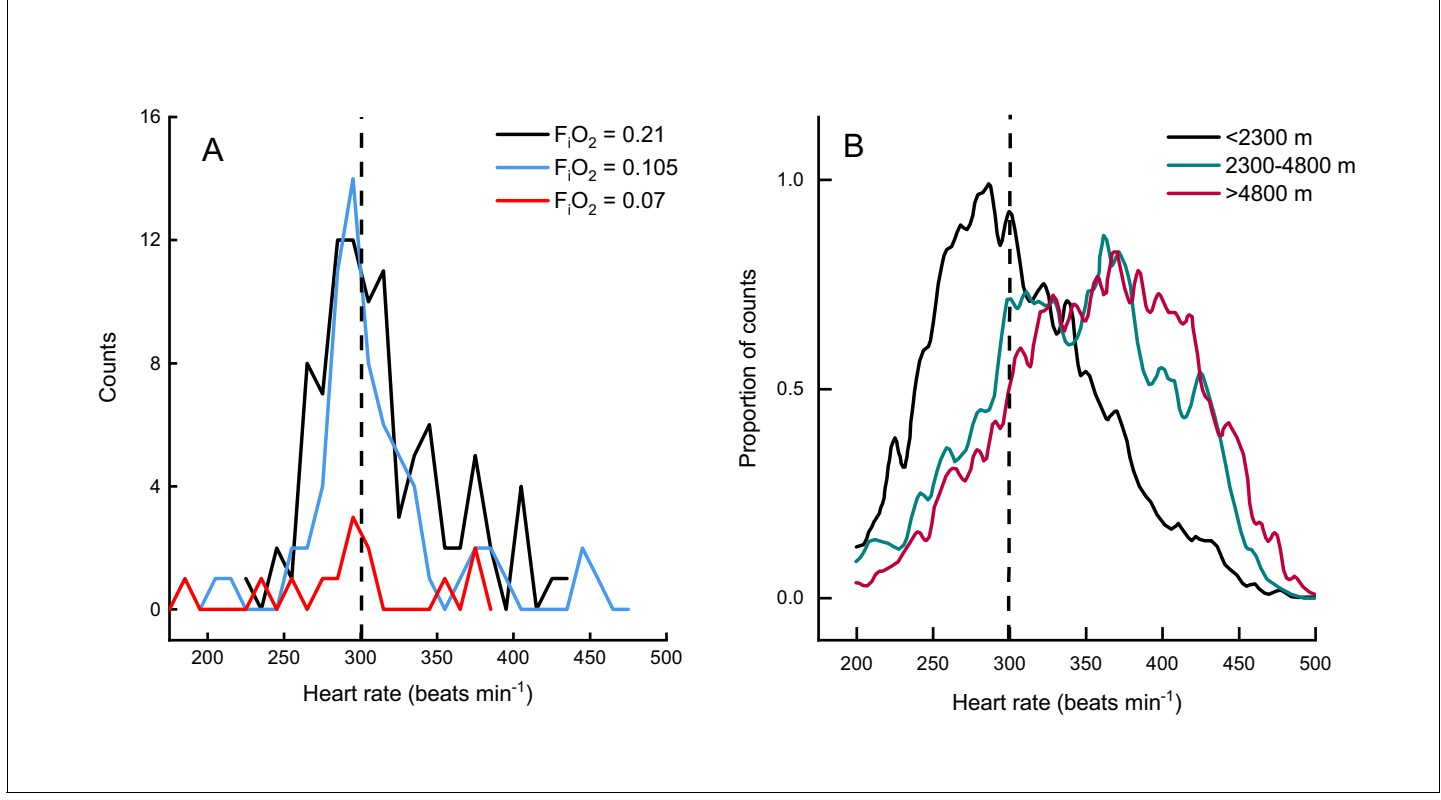

**Figure 2.** Heart rate during flights of bar-headed geese. Line histogram distribution of heart rate measurements during flight for the present study (**A**) and measurements taken from wild, migrating birds by *Bishop et al. (2015)* (**B**). Dashed line shown to indicate 300 beats per minute in each plot (for aid in visual comparison only). Note that only one bird flew consistently in severe hypoxia (red trace in panel **A**).
DOI: https://doi.org/10.7554/eLife.44986.007

normoxia (85.8 ± 10.8 mmHg) was significantly higher (Q = 7.079, p=0.006) than arterial $P_{O_2}$ in moderate hypoxia (47.0 ± 4.3 mmHg), and severe hypoxia (36.3 ± 2.7 mmHg, Q = 9.537, p=0.001).

Example $P_{O_2}$ recordings during normoxic and hypoxic flights are shown in *Figure 4*. Note the rapid rise in mixed venous temperature immediately after landing, followed by a rapid recooling, and slow rewarming phase. This was consistently observed during flight recovery.

We measured blood gas variables in resting birds during periods separate from flight trials (see rest data in *Supplementary files 2* and *3*). There was a significant effect of oxygen level on venous $P_{O_2}$ at rest ($F_{2, 17.33}$=27.775, p<0.0001). Both moderate hypoxia (*EMM*=28.96 ± 3.18 mmHg; t=−5.579, p=0.0001) and severe hypoxia (*EMM*=23.36 ± 3.44 mmHg; t=7.001, p<0.0001) were significantly lower than normoxia (*EMM*=46.51 ± 3.18 mmHg), but not significantly different from each other (t=-1.694, p=0.3238). Venous temperature did not differ significantly between $O_2$ levels at rest ($F_{2, 4}$=3.2428, p=0.1455).

## Discussion

Six of seven captive birds (born and raised at sea level) that were successfully trained to fly in the wind tunnel were willing to fly in moderate hypoxia equivalent to the altitudes at which their wild conspecifics migrate (~5,500 m). Three bar-headed geese also flew in severely hypoxic conditions [equivalent to an altitude of roughly 9,000 m (0.07 $F_iO_2$)], at least for the short duration of flights in this study. We found that the primary contribution to increased gas transport between rest and flight in both normoxia and hypoxia was from increases in the estimated $O_2$ pulse (between 6 and 8-fold; inferred from the calculated $CO_2$ pulse). Increases in heart rate contributed less (between 2 and 3-fold), with large variations in heart rate at any level of $CO_2$ production and vice versa. The duration

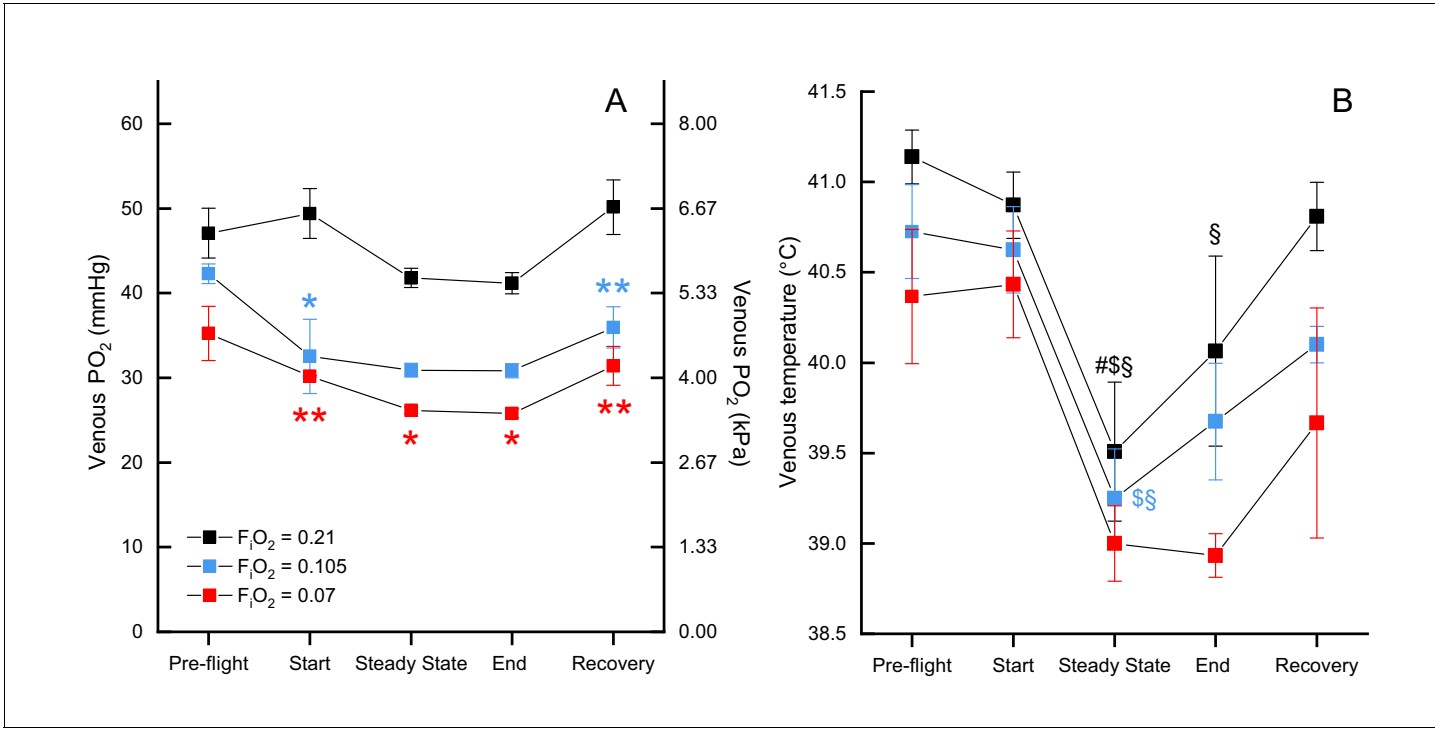

**Figure 3.** Mixed venous Po₂ and temperature during flight. Mixed venous Po₂ (**A**) and temperature (**B**) in FiO₂=0.21 (black, n=4 birds, 13 flights), FiO₂=0.105 (blue, n=2 birds, 6 flights), and FiO₂=0.07 (red, n=1 bird, 3 flights). Time points shown along x-axis: 'pre-flight' is steady state before flight begins, 'start' at the start of the flight, 'steady state' is steady state in flight, 'end' at the end of the flight, 'recovery' is steady state after the bird lands. Asterisks indicate significant difference from normoxia (* indicates p<0.05; ** indicates p<0.01; *** indicates p<0.001, § indicates difference from pre-flight value, # indicates difference from recovery value, and $ indicates difference from start value).
DOI: https://doi.org/10.7554/eLife.44986.008

of experimental flights and heart rate were unaffected by moderate hypoxia; reductions in $O_2$ availability were largely matched by reductions in metabolic rate.

## Methodological considerations

Flights in our study were relatively short compared to wild flights by conspecifics at high altitude (mean duration for flight over the Himalayas is 8 hr; *Hawkes et al., 2013*) but are comparable to those of a previous wind tunnel study. *Ward et al. (2002)* experienced the same difficulty (*Pers. Comm.*), with only two of their five bar-headed geese achieving flight in the wind tunnel, and flights remaining relatively short. Given that birds underwent considerable training, including outdoor flights, and that wind tunnel flights were short even in normoxia, it would appear that the birds were reluctant to fly for long once instrumented in the conditions of the wind tunnel. Flow turbulence in the tunnel, the presence of the experimenters and the presence of the mask and tubing all will have increased flight costs and may have contributed to this (*Hedenström and Lindström, 2017*). Although wing-beat frequencies of our birds were higher than those of bar-headed geese in the wild (*Bishop et al., 2015*), values were similar between normoxic vs. hypoxic and instrumented vs. uninstrumented flights (*Supplementary file 4*; *Whale, 2012*). Despite possible instrumentation effects or the short flight durations, flights were repeatable, of similar length under all conditions, and most importantly, produced stable levels of the measured variables, allowing us to make robust comparisons between flight in normoxia vs. hypoxia, thus examining the effects of hypoxia on flight physiology under similar conditions. Determining how these results relate to the multi-hour migratory flights of this species at high altitude will require further work measuring physiological variables in the wild, or during longer flights in both normobaric and hypobaric conditions.

It was challenging to obtain reliable measures of $\dot{V}_{O2}$ during flight in hypoxia, likely because of small fluctuations in gas mixing, given the dynamics of flight in the wind tunnel while wearing the

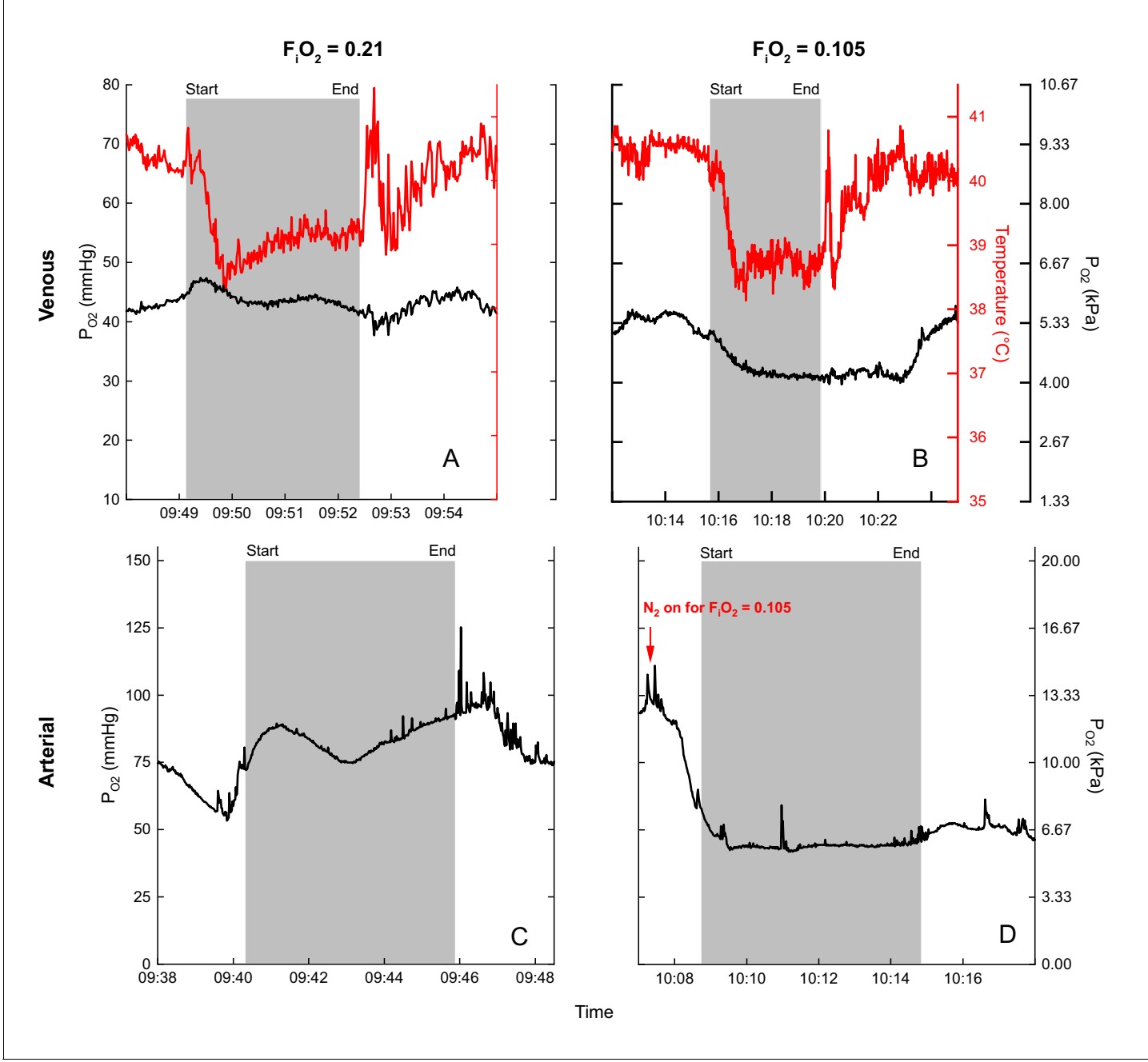

**Figure 4.** Examples of blood $P_{O2}$ (black) and temperature (red) recordings during flight (gray shaded area) in normoxia (venous (**A**), arterial (**C**)) and hypoxia (venous (**B**), arterial (**D**)) for bird 45. Flight duration of (**A**) 3.3 minutes, (**B**) 4.2 minutes, (**C**) 5.7 minutes, and (**D**) 5.5 minutes.
DOI: https://doi.org/10.7554/eLife.44986.009

mask. Similar problems were encountered in a previous study (*Hawkes et al., 2014*). While small minute-to-minute fluctuations in $F_iO_2$ will average out and not alter the hypoxic challenge to the bird, they do have a destabilizing effect on the calculation of $\dot{V}_{O2}$. Because incurrent $CO_2$ levels remained close to zero throughout, any increase in $CO_2$ must come from the bird and therefore our $\dot{V}_{CO2}$ data were considered robust. Because the RER averaged 0.988 ± 0.01 during flight in normoxia and flight durations between normoxia and moderate hypoxia were not significantly different, we have made the assumption that $\dot{V}_{CO2}$ and $\dot{V}_{O2}$ can be used interchangeably under this condition. As opposed to indicating carbohydrate use during flight, an RER near 1 may reflect hyperventilatory

$CO_2$ loss. We would expect during longer flights that RER would fall close to 0.7, assuming the birds are preferentially metabolizing lipids. This is supported by our data as RER falls to $0.921 \pm 0.02$ for flights longer than six minutes.

## Comparisons to existing data

Heart rate and metabolic rate of bar-headed geese at rest in normoxia in this study were remarkably similar to those obtained by *Ward et al. (2002)*, as was the mean respiratory exchange ratio (RER). However, wingbeat frequencies measured during flight in normoxia in the previous study were lower, as was $\dot{V}_{CO2}$ (by approximately 29%). Heart rates during flight, however, were lower in the current study suggesting that our birds were working harder but were employing larger increases in cardiac output and/or pulmonary exchange (*Figure 1B*). The discrepancy in heart rates may also be due to methodological differences, as we found it necessary to visually verify each heart rate peak while Ward et al. relied on periodic averages.

Birds in the present study flew at a slightly lower range of flight speeds compared to birds in the wild [12.5 to 15.0 m s$^{-1}$ versus 17.1 m s$^{-1}$ for wild bar-headed geese migrating at <1,000 meters altitude (*Hawkes et al., 2013*). The 2.5-fold difference in heart rate measured between birds at rest and in-flight in the present study is also similar to that of wild birds flying at low altitudes (*Bishop et al., 2015*) (*Figure 2*). Bishop et al. documented an increase in heart rate with increasing altitude. As heart rate in the current study was unaffected by hypoxia (*Table 1*), this suggests that the increases in heart rate measured in wild bar headed geese migrating at altitudes above 2,300 meters may be a consequence of flight dynamics in hypobaria, rather than hypoxia. Finally, heart rate was highly variable at any level of $CO_2$ production and vice versa. This was also the case for the relationship between heart rate and wing-beat frequency in wild birds, although mean values were well correlated (*Bishop et al., 2015*). *Ward et al. (2002)* also concluded that their wind tunnel data could not be used directly to calculate the metabolic rate of wild migratory geese from measurements of heart rate alone.

## Effects of hypoxia

In hypoxia both at rest and preflight in the wind tunnel, $\dot{V}_{CO2}$ fell by 22 and 26% for $F_iO_2$=0.105 and by 10 and 29% for $F_iO_2$=0.07. In the one bird for which we have data at all $O_2$ levels, arterial $P_{O2}$ fell to $56.5 \pm 5.4$ and $36.7 \pm 0.54$ mmHg preflight for $F_iO_2$=0.105 and $F_iO_2$=0.07, respectively. Based on the data from *Meir and Milsom (2013)* and assuming a body temperature of 41°C and an arterial pH of 7.4, this would lead to a fall in $O_2$ saturation pre-flight from around 92% (0.21 $F_iO_2$) to 84% (0.105 $F_iO_2$) and 67% (0.07 $F_iO_2$), roughly equivalent to the decrease in metabolic rate.

For birds flying in $F_iO_2$ = 0.105, $\dot{V}_{CO2}$ was 16% lower than in birds flying in normoxia. Heart rates in moderate hypoxia were not significantly different from those under any state (rest, pre-flight, flight) in normoxia, while the estimated $O_2$ pulse decreased in proportion to the $\dot{V}_{CO2}$. The greater than 8-fold increase in $O_2$ pulse from rest to flight in normoxia was maintained in moderate hypoxia. Thus the moderately hypoxic birds appear to have met the hypoxic challenge by a combination of a reduced metabolism, maintaining heart rate, and maintaining the increase in $CO_2$/estimated $O_2$ pulse.

For the one bird for which we have adequate data flying in $F_iO_2$ = 0.07, $\dot{V}_{CO2}$ was 20% lower under this severe hypoxic condition than in normoxia. Again, heart rates were not significantly different when flying in hypoxia. This bird did have a higher heart rate ($333.6 \pm 11$ beats min$^{-1}$) despite the lower $\dot{V}_{CO2}$ ($157.4 \pm 8.4$ ml $CO_2$ kg$^{-1}$ min$^{-1}$) in normoxia than the other birds, likely also contributing to its exceptional performance.

The reduction in metabolism in hypoxia observed in the current study could represent $O_2$ limitation, selective suppression of metabolism to specific tissues or increased efficiency of flight pattern and thus $O_2$ utilization. Alternatively, the reduction measured in metabolic rate could be concordant with the onset of anaerobic metabolism. Although we cannot reject this possibility as lactate was not measured in this study, we consider it unlikely as there was no sign of an oxygen limitation, because: 1) the birds could still increase $\dot{V}_{CO2}$ by 14 to 23-fold during flight, 2) reductions in metabolic rate also occurred under rest and preflight conditions, and 3) the birds sustained flights of similar durations at constant levels of arterial $P_{O2}$. It is quite possible that while flying under the more metabolically challenging conditions of hypoxia, the birds are minimizing energy supply to less essential

processes (e.g. digestion, birds are known to undergo atrophy of gut tissue prior to migration; *McWilliams et al., 2004*; *Piersma and Gill, 1998*), or that they may be altering their flight behavior and biomechanics to fly with maximal efficiency. This is supported by the metabolic data as the individual minimum metabolic rates (the lowest steady state $\dot{V}_{CO_2}$ of all flights for each bird) were not different between normoxia and moderate hypoxia. Only the overall average metabolic rate differs, indicating that birds may employ more or less efficient flight strategies in normoxia, but shift towards using only the most efficient strategies when oxygen limited. Wing-beat frequencies of bar-headed geese in this study were similar in both normoxia and hypoxia. This is consistent with results from both ruby-throated hummingbirds (*Archilochus colubris*) and the South American hummingbird (*Colibri coruscans*), a montane species capable of hovering at altitudes over 6000m (*Chai and Dudley, 1996*; *Berger, 1974*). Despite a constant wing beat frequency, flight biomechanics of the geese in our study were altered in response to hypoxia, with increased upstroke duration (T) and decreased upstroke wingtip speed (Utip), upstroke plane amplitude (FSP), and mid-upstroke angle of inclination (a) (*Supplementary file 4*; *Whale, 2012*) As the downstroke produces the majority of lift and all forward thrust, by increasing the ratio of the duration of upstroke to downstroke, the duration of activation of the pectoralis major muscle group is decreased (responsible for the majority of downstroke power). We therefore hypothesize that bar-headed geese reduce oxygen demand in hypoxic flight by limiting oxygen supply to less essential metabolic processes and/or maximizing the mechanical efficiency of flight.

## Vascular $P_{O_2}$ and temperature

We obtained the first measurements of arterial and venous $P_{O_2}$ and temperature records in this species, and that of any equivalently sized bird, during flight. Both mixed venous and arterial $P_{O_2}$ values decreased progressively with decreasing levels of $F_iO_2$, as expected. In general, levels of arterial $P_{O_2}$ were maintained throughout flights (*Figure 4*) although there was some variability in individual flights. Mixed venous $P_{O_2}$, on the other hand, tended to decrease during the initial portion (first minute) of flights in hypoxia (*Figure 3* and *Figure 4*), indicative of increased tissue $O_2$ extraction.

The arterial $P_{O_2}$ of geese flying at 0.105 $F_iO_2$ was similar to that of geese running on a treadmill in a previous study at 0.07 $F_iO_2$ (*Figure 4*; *Hawkes et al., 2014*). When directly comparing at the same level of hypoxia (0.07 $F_iO_2$ for both studies), arterial $P_{O_2}$ during flight was about 20% lower than while running *Hawkes et al. (2014)*. Arterial values in the range measured in 0.07 $F_iO_2$ are strikingly low (*Supplementary files 1* and *3*), particularly given the need to support the metabolically costly activity of flight. Interestingly, these values are equivalent to the mean minimum arterial $P_{O_2}$ values obtained near the end of dives in elephant seals, and are similar to the range exhibited by diving emperor penguins (*Ponganis et al., 2007*; *Fedak et al., 1981*). These $P_{O_2}$ values correspond to quite different blood oxygen saturation ($S_{O_2}$) values between these species, however, due to the inherent differences between the flight environment and breath-hold diving and their subsequent effects on the $O_2$-Hb dissociation curve. For example, the associated hyperventilation (decreased $CO_2$) and decrease in temperature (below) in the flying goose correspond to a much higher arterial $O_2$ content for the same low levels of $P_{O_2}$ experienced between these species (*Meir and Milsom, 2013*).

As near complete utilization of the available $O_2$ store (venous $P_{O_2}$ values near zero at the end of dives) certainly contributes to the success of elite divers like elephant seals and emperor penguins (*Ponganis et al., 2007*; *Meir et al., 2009*), the capacity to effectively maximize $O_2$ resources in the $O_2$-limited environment of high altitude flight would also afford a distinct advantage. With venous $O_2$ values decreasing to only around 25–30 mmHg in the present study, even under extreme hypoxia, these high fliers may yet retain a venous $O_2$ reserve, also suggesting that these birds were not $O_2$ limited in hypoxic flight. Venous $P_{O_2}$ values as low as 2–10 mmHg have been reported during dives in elite divers like elephant seals and emperor penguins (*Ponganis et al., 2007*; *Meir et al., 2009*), or in hypoxemic extremes of race horses performing strenuous exercise (*Manohar et al., 2001*; *Butler et al., 1993*; *Bayly et al., 1989*). Hopefully, further gains made in the field of bio-logging systems directly measuring $P_{O_2}$ or $S_{O_2}$ will elucidate these variables in wild, migrating birds in the future.

One of our key findings was the consistent fall in venous temperature during flight (*Figure 3* and *Figure 4*). As the temperature probes were inserted through the jugular vein and advanced to the

level of the heart, these records should reflect true mixed venous temperature. The decrease in temperature may indicate cooling of blood flowing through the buccal/pharyngeal cavity *via* evaporative water loss from the respiratory passages and/or restriction of blood flow to the gut. As it has been demonstrated that the blood of the bar-headed goose has high thermal sensitivity (*Meir and Milsom, 2013*), this drop in temperature could enhance $O_2$ loading considerably as the blood subsequently cools in the lung (*Meir and Milsom, 2013*). For example, when converting venous $P_{O_2}$ values (*Figures 3*,*4*, *Supplementary file 2*) into Hb-$O_2$ saturation (*Meir and Milsom, 2013*), the corresponding temperature drop in hypoxia results in a substantial increase in $O_2$ content, indicating an even larger venous reserve than that inferred from the $P_{O_2}$ values alone. As blood travels away from the lung toward the exercising tissue, it would be expected to warm, enhancing $O_2$ unloading. Temperature profiles also reveal a transient spike in temperature immediately following each flight, perhaps due to a release of warm blood from exercising muscle or other areas. This characteristic spike is followed by a second bout of cooling, and then a slow warming to levels at rest (*Figure 4*). Measurements of temperature at the lung and at the muscle in wild, migrating birds would help determine if modulating blood temperature might increase oxygen flux during flight.

## Perspectives and significance

Previous studies have documented wild, migrating geese flying regularly between 5,000–6,000 m above sea-level and as high as 7,290 m (*Bishop et al., 2015*; *Hawkes et al., 2013*), with earlier anecdotal reports suggesting that these birds may even fly as high as the summit of Mt. Makalu (8,485 m; *Swan, 1961*). Based on the ability of geese flying under hypoxic conditions in the present study in a wind tunnel, we believe that, although these geese routinely use lower mountain passes during their migration, their suite of physiological adaptations could support flight even at extreme altitudes. We suggest that this would largely be possible *via* a reduction in metabolism in hypoxia, while maintaining the heart rate and relative-increase in $O_2$ pulse also measured in flight in normoxia. Interestingly, blood temperature dynamics may also play a critical role in enhancing $O_2$ loading in this species during its exceptional migration.

# Materials and methods

All experiments were conducted according to UBC Animal Care Committee protocols A14-0051 and A14-0136 under the guidelines of the Canadian Council on Animal Care.

## Imprinting and training

To facilitate wind tunnel training, geese were imprinted on the experimenters. Briefly, bar-headed goose (*Anser indicus*) eggs were obtained from the Sylvan Heights Bird Park (Scotland Neck, North Carolina). Geese (twelve bar-headed geese in 2010, seven in 2011) were imprinted on a human foster parent (J.U.M. in 2010 and J.M.Y. in 2011) during the first several weeks at the waterfowl park and then transported to Vancouver, B.C., Canada (in accordance with U.S. Department of Agriculture Animal and Plant Health Inspection Services and Canadian Food Inspection Agency protocols/inspections). They were then housed in the University of British Columbia Animal Care Facility with constant access to water (small ponds) and *ad libitum* mixed grain pellets supplemented with lettuce for the duration of the project. Birds were familiarized with dummy respirometry masks and backpack systems soon after hatching. Because the wind tunnel was undergoing repair when the first year's (2010) birds fledged, they were initially taken on outdoor training flights alongside their foster parent on a bicycle, and later on a motor scooter, to facilitate development of flight muscle and physiological capacity (*Video 1* and *Figure 5—figure supplement 1*).

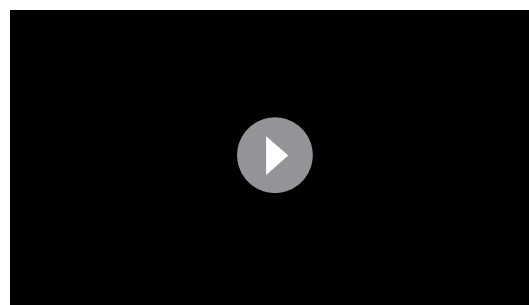

**Video 1.** Flight training with goose and foster parent (JUM) on a motor scooter, undertaken as the wind tunnel was undergoing repair at the time of fledging.
DOI: https://doi.org/10.7554/eLife.44986.010

## Wind tunnel flights

Geese were flown in the University of British Columbia (UBC) Department of Mechanical Engineering's boundary layer 30 m open-circuit wind tunnel (http://mech.ubc.ca/alumni/aerolab/facilities/). Airspeed in the test section (1.6 m high x 2.5 m wide x 23.6 m long) between 3 to 20 m s$^{-1}$ was calibrated using a pitot tube system built into the tunnel. Experimental flights took place primarily during times that corresponded to spring and fall migration of wild bar-headed geese (Jan. 2011-Nov. 2012). The range of wind tunnel flight speeds selected was similar to that measured during natural migratory flight (14 to 21 m s$^{-1}$; *Hawkes et al., 2013*; *Hawkes et al., 2011*) and the speed selected for each individual was that which allowed steady, stationary, and prolonged flight. Three birds flew at 12.5 m s$^{-1}$, one bird at 13.75 m s$^{-1}$, and three birds at 15 m s$^{-1}$. As the wind tunnel is an open-loop system continuously drawing in outside air while operating, temperature in the wind tunnel was equivalent to ambient local outdoor temperature (range: 3–21°C) for each flight. The foster parent stood against the wall at the front of the flight section of the wind tunnel to encourage the bird to sustain flight. Another investigator initially lifted the bird into the air stream from behind, then supported the tubing running from the mask to the data acquisition system, holding them 2–3 feet above and behind the bird to allow free movement of the flying bird (*Figure 5* and *Video 2*). Seven bar-headed geese (2.21 ± 0.26 kg) managed steady, stationary, and prolonged flight in the wind tunnel while fully instrumented.

## Physiological measurements

We measured heart rate ($f_H$), the rate of oxygen consumption ($\dot{V}_{O_2}$) and the rate of $CO_2$ production ($\dot{V}_{CO_2}$) under conditions at rest and during flight in bar-headed geese in both normoxia and two levels of hypoxia (moderate: 0.105 and severe: 0.07 $F_iO_2$ equivalent to altitudes of roughly 5,500 m and 9,000 m respectively). Subcutaneous electrodes were inserted dorsally proximal to the spine: one at the level of the axilla and the second near the pelvis. The electrodes were connected to a custom-built 3-channel $P_{O2}$/temperature/electrocardiogram (ECG) digital recorder (UFI, Morro Bay, CA, USA) (*Meir et al., 2008*; *Ponganis et al., 2007*; *Fedak et al., 1981*; *Meir et al., 2009*; *Ponganis et al., 2009*), which sampled at 100 Hz (*Meir et al., 2008*; *Ponganis et al., 2007*). $\dot{V}_{O_2}$ and $\dot{V}_{CO_2}$ were measured using mask respirometry. Two ports in the mask drew ambient, normoxic air in and over the nares via space at the top of the mask and introduced oxygen-free nitrogen from behind the nares such that it mixed with the ambient air to provide a hypoxic gas mix flowing over the nares for the goose to breathe (0.105 and 0.07 $F_iO_2$). The airflow rate through the mask was 70 l min$^{-1}$ during flight and 10 l min$^{-1}$ at rest, which was sufficient to prevent any leakage from the mask, tested using nitrogen dilutions [16]. A subsample (200 ml min$^{-1}$) of air from the mask was drawn through a Sable Systems Field Metabolic System (FMS) (Sable Systems, Las Vegas, NV, USA), via a desiccant membrane dryer (AEI Technologies, Pittsburgh, PA, USA, *Figure 5* and *Video 2*), which was calibrated at the start and end of each trial.

Masks were custom-made on a Plaster of Paris cast of the head of a deceased bar-headed goose using heat-moldable dental mouth-guard compound (Thermo-Forming Material, Clear-Mouthguard,. 040', Henry Schein, Canada), which was softened with a heating gun and stretched over the cast to create a light-weight, form-fitted mask that could be secured with a thin elastic strap below the base of the skull. The mask covered the beak and forehead of the goose but did not cover the eyes.

The tubes sampling air and delivering nitrogen to the mask exited the tunnel to the respirometry set-up at an access point (*Figure 5*). Gas (air or the hypoxic gas mixture) was drawn from the mask by a dry rotary vane vacuum pump (4.5 cfm, 115 VAC, Cole-Parmer, Montreal, Quebec, CA, equipped with vacuum gauge and vacuum relief valve) controlled with a digital mass flow controller (Sierra Instruments Smart-Trak100, 0–200 SLPM, Accuracy: + 1% of full scale, BG Controls, Port Coquitlam, BC, Canada). A 10 μM nylon net filter (Millipore, Billerica, MA, USA) was used to prevent dust, down, or debris from entering the flow controller. Rotameters (Acrylic Flowmeter, FL-2042, 3 to 30 l min$^{-1}$, FL-2044, 10 to 100 l min$^{-1}$, Omega, Laval, QC, Canada) were used to generate flow rates of oxygen-free $N_2$ sufficient to produce 0.105 and 0.07 $F_iO_2$ in the respirometry mask for both flight and rest, using the Plaster of Paris goose head mold for the calibrations. The gas analyzer was calibrated to account for sensor drift using: 1) two point calibration for $CO_2$, 0% and 1.0% $CO_2$ balance air (Praxair Canada, Scarborough, ON, Canada); 2) a single point calibration for $O_2$ at a

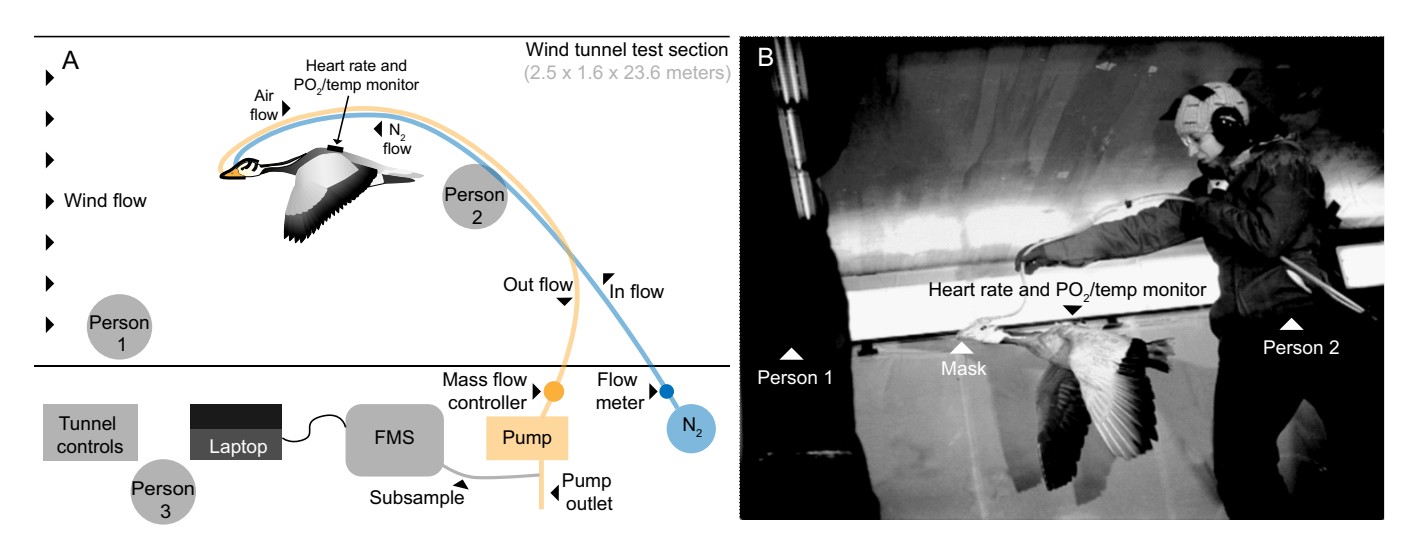

**Figure 5.** Flight tunnel experimental set up. (**A**) Schematic and (**B**) photo showing the set up in the wind tunnel. The goose flew in center of wind tunnel test section while person one encouraged flight and person two supported tubing. Tubes ran from mask out of the tunnel, one introducing a calibrated amount of dry nitrogen into the mask, and the other pulling from the mask by way of an air pump. A subsample of the outflow was pulled into the field metabolic system (FMS). Person three operated the tunnel and equipment.

DOI: https://doi.org/10.7554/eLife.44986.011

The following figure supplement is available for figure 5:

**Figure supplement 1.** Flight training with goose and foster parent (JUM) on a motor scooter, undertaken as the wind tunnel was undergoing repair at the time of fledging.

DOI: https://doi.org/10.7554/eLife.44986.012

baseline of 20.95% for dried room air at experimental flow rates since zero is extremely stable (*Fedak et al., 1981*).

## Blood gases

To determine whether arterial blood oxygen was maintained in flight, and the extent of the venous blood $O_2$ reserve remaining following tissue $O_2$ extraction, arterial and venous blood $P_{O_2}$ and temperature were measured at rest and during flight using intravascular $P_{O_2}$ electrodes (Licox $P_{O_2}$ microprobe, Canada Microsurgical Ltd., Burlington, ON, Canada) and thermistors (Yellow Springs Instruments model 555, Fisher Scientific, Edmonton, AB, Canada), introduced using aseptic surgical technique under general isoflurane anesthesia, with meloxicam as an analgesic. Only one site, either arterial or venous, was targeted per surgery and subsequent flights (n=5 birds).

For venous deployments, $P_{O_2}$ electrodes and thermistors were inserted percutaneously via the right jugular vein using a peel-away catheter over needle (Arrow 15 Ga, Teleflex Medical, Markham, Ontario, Canada; similar to methods described in *Meir et al., 2008*; *Ponganis et al., 2007*; *Meir et al., 2009*; *Ponganis et al., 2009*). Electrodes were inserted to lie close to the heart to sample mixed venous blood (ranging from 9 to 13.5 cm from insertion site to cannula tip, depending on the bird and insertion site). For arterial deployments, $P_{O_2}$ electrodes were inserted in the aorta via the carotid artery using

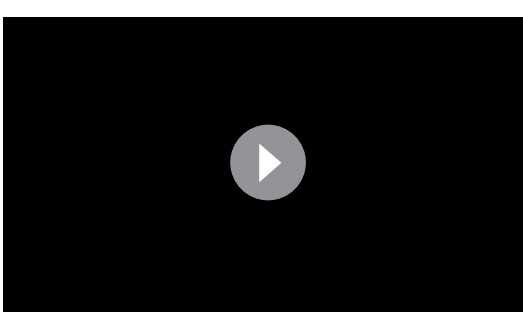

**Video 2.** Goose flying in tunnel during $F_iO_2 = 0.105$ experiment. Person one can be seen to the left of the screen, person two supported the tubes, and person three operated the experimental hardware and wind tunnel. Filmed at 125 frames per second, shown here at 7.5 frames per second playback. Video credit: J. Whale.
DOI: https://doi.org/10.7554/eLife.44986.013

peel-away catheters (3.5 FR Peel-Away Denny Sheath Introducer Set, Cook Medical Inc, Blooming-ton, IN, USA or Arrow 17 Ga, Teleflex Medical) after exposing the vessel via a shallow incision. The thermistor could not be deployed simultaneously with the arterial $P_{O_2}$ electrode due to aortic size.

Cannulae were coiled and secured with a purse string suture at the insertion site, and covered with medical tape. Bioclusive transparent film dressing (Henry Schein, Melville, NY, USA) was placed over the insertion site of the electrodes and Elastinet stocking placed over the neck to protect the insertion site, secured by Bioclusive at each end. Animals recovered overnight from surgical proce-dures before experimental sessions in the wind tunnel were conducted. At the start of each experi-ment the $P_{O_2}$ electrode and thermistor were attached to the custom-built recorder (see *Ponganis et al., 2007*; *Meir et al., 2009*; *Ponganis et al., 2009*). The $P_{O_2}$ electrode and thermistor, calibration procedures and verification testing have been described previously (*Meir et al., 2008*; *Ponganis et al., 2007*; *Meir et al., 2009*; *Ponganis et al., 2009*). At the end of the experiments the cannulae were removed and the animals inspected by veterinary surgeons and recovered in outdoor aviaries.

## Physiological measurements and data analysis

$\dot{V}_{O_2}$ was calculated as the difference between the fractional concentrations of $O_2$ in dry inspired ($F_iO_2$) and expired ($F_eO_2$) air (*Lighton, 2008*) as follows:

$$\dot{V}_{O2} = \dot{V}_{STPD}(F_iO_2 - F_eO_2) - \frac{F_iO_2(F_eCO_2 - F_iCO_2)}{(1 - F_iO_2)}$$

Where $\dot{V}_{STPD}$ is the flow rate of the gas being drawn from the mask. $\dot{V}_{CO2}$ was calculated as:

$$\dot{V}_{CO2} = \dot{V}_{STPD}(F_eCO_2 - F_iCO_2) + \frac{F_iCO_2(F_iO_2 - F_eO_2)}{(1 + F_iCO_2)}$$

The start and end of each flight was determined from the data trace by an obvious change in $CO_2$ production. Data were used only if a stable plateau in $CO_2$ production had been reached. $F_eO_2$ and $F_eCO_2$ were determined as the average across this entire portion of the trace. Stable data were obtained under all conditions for $\dot{V}_{CO2}$, however it was not possible to gather reliable $\dot{V}_{O2}$ data in hypoxia (as in other studies: *Hawkes et al., 2014*). Therefore, $\dot{V}_{O2}$ data are not reported for flights in hypoxia. Respiratory exchange ratios (RER) were calculated by dividing $\dot{V}_{CO2}$ by $\dot{V}_{O2}$ and could therefore only be calculated for data collected in normoxia. ECG data were analyzed using peak detection software to automatically mark R-waves (all data were then visually verified). Heart rate was counted as the mean during the same period used above for respirometry analysis. All $P_{O_2}$ values were temperature corrected for construction of $P_{O_2}$ profiles as previously described (*Ponganis et al., 2007*). For arterial deployments in which temperature data could not be obtained, temperature was assumed to be stable at baseline body temperature (41˚C).

## Calculations and statistical analysis

$CO_2$ pulse (an indication of how much $CO_2$ is transported in the blood by each heartbeat) was calcu-lated by dividing the rate of $CO_2$ production ($\dot{V}_{CO2}$) by heart rate ($f_H$):

$$CO_2\ pulse = \frac{\dot{V}_{CO2}}{f_H}$$

According to the Fick equation, this is equivalent to the product of stroke volume (SV) and the arterial-venous $CO_2$ difference ($Ca_{CO2} - Cv_{CO2}$).

$$CO_2\ pulse = \frac{\dot{V}_{CO2}}{f_H} = SV\ (Ca_{CO2} - Cv_{CO2})$$

As stroke volume (SV) and the arterial-venous $CO_2$ difference were not measured in our study, however, our data cannot differentiate between the two. $O_2$ pulse was also estimated during nor-moxic flights to calculate putative $\dot{V}_{O2}$ during hypoxic flights, assuming an RER of 1.0 to convert $\dot{V}_{CO2}$ into $\dot{V}_{O2}$.

Data were plotted using Origin2016 software (OriginLab, Northampton, MA, USA). To statistically investigate our data while accounting for repeated measures among individual birds we used a linear mixed model approach with a random effect of the individual bird. This accounts for within-subject variance by assigning each bird an individual intercept. We generated individual models for each dependent variable (duration, RER, $\dot{V}_{CO2}$, heart rate, $CO_2$ pulse, blood $PO_2$, venous temperature) and compared main effects of oxygen level (partial pressure of oxygen) and activity or time point, as well as the interaction (oxygen level*activity) and compared estimated marginal means post-hoc assuming significance at $p < 0.05$. We used the afex package in RStudio (R version 3.5.1) for generating the models, the emmeans package for post-hoc comparisons with Bonferroni adjustment where appropriate, and calculated the adjusted intraclass correlation coefficient (ICC) by dividing the variance of the random intercept by the sum of the random effect variances (a value closer to 1 indicates a greater effect of the individual bird). We report estimated marginal means (*EMM*) in the results where indicated, and descriptive statistics in *Table 1*.

For comparisons involving one individual bird, dependent variables were compared using Sigma-Plot software (Systat Software Inc, San Jose, CA, USA) based on oxygen level, activity, or time point using t-tests or one-way ANOVAs with post-hoc Tukey tests where appropriate. When normality of data was not achieved, groups were compared using Kruskal-Wallis one-way ANOVA on ranks with post-hoc Dunn's test assuming significance at $p < 0.05$. R script and data files (including source data for figures, although figures were not generated in R) were deposited in Dryad (doi:10.5061/dryad. fg80hp6).

## Acknowledgements

This work would not have been possible without the tireless efforts of the numerous UBC undergraduate volunteers that assisted with animal husbandry, training, and data collection (in particular, Erin Erskine, Michelle Reichert, Anthony Pang, Alice Kuan, Judy Cha, Deanna White, Christine Yeung, Winnie Cheung and Nici Darychuk). We are indebted to Gordon Gray and the staff of the UBC Centre for Comparative Medicine for their tireless support and expertise in animal care. We thank Charles Bishop, Sally Ward, and Pat Butler for helpful suggestions and correspondence during the experimental design and training phases and for critical feedback through numerous discussions. We are grateful to Sheldon Green, Peter Ostafichuk and students of the UBC Dept. of Mechanical Engineering for use of the wind tunnel; Marty Loughry and Tom Wright of UFI for design and construction of the recorders; Bob Shadwick for use of his rad scooter and transport van; Yvonne Dzal for her mad chauffeur skills; James Whale for flight kinematic data and video; Graham Scott for manuscript review; and Erika Hale for assistance with statistics. We encourage anyone who is interested in waterfowl to visit the Sylvan Heights Bird Park and thank Mike and Ali Lubbock, Nick Hill and their incredible personnel for their invaluable assistance and hospitality in acquiring, imprinting and raising the geese. We thank the reviewers of this manuscript for their valuable comments and suggestions.

## Additional information

### Funding

| Funder | Grant reference number | Author |
| --- | --- | --- |
| National Science Foundation | International Research Fellowship Program (IRFP) post-doctoral fellowship (OISE-0855669) | Jessica U Meir |
| Natural Sciences and Engineering Research Council of Canada | | William K Milsom |
| National Science Foundation | RAPID Grant (OISE-1154099) | Jessica U Meir |

The funders had no role in study design, data collection and interpretation, or the decision to submit the work for publication.

## Author contributions
Jessica U Meir, Conceptualization, Resources, Data curation, Software, Formal analysis, Supervision, Funding acquisition, Validation, Investigation, Visualization, Methodology, Writing—original draft, Project administration, Writing—review and editing; Julia M York, Data curation, Software, Formal analysis, Investigation, Visualization, Methodology, Writing—original draft, Writing—review and editing; Bev A Chua, Wilhelmina Jardine, Investigation, Methodology; Lucy A Hawkes, Methodology, Writing—review and editing; William K Milsom, Conceptualization, Resources, Supervision, Funding acquisition, Visualization, Methodology, Writing—original draft, Project administration, Writing—review and editing

## Author ORCIDs
Jessica U Meir (iD) https://orcid.org/0000-0002-4215-4302
Julia M York (iD) https://orcid.org/0000-0003-4947-0591

## Ethics
Human subjects: Although the subjects in this study were not human, the investigators do appear in supplementary files (photographs). Consent has been obtained by all subjects in the photo and videos used in this manuscript.
Animal experimentation: All experiments were conducted according to UBC Animal Care Committee protocols A14-0051 and A14-0136 under the guidelines of the Canadian Council on Animal Care.

## Decision letter and Author response
Decision letter https://doi.org/10.7554/eLife.44986.022
Author response https://doi.org/10.7554/eLife.44986.023

# Additional files

## Supplementary files
• Supplementary file 1. Physiological measurements from bird 45 only. Values are mean ± SEM. Asterisks indicate significant difference from normoxia (ANOVA; * indicates $p<0.05$; ** indicates $p<0.01$; *** indicates $p<0.001$).
DOI: https://doi.org/10.7554/eLife.44986.014

• Supplementary file 2. Mixed venous $P_{O_2}$ and metabolic data collected from the subset of flights during which mixed venous $P_{O_2}$ was measured. Values are mean ± SEM. Asterisks indicate significant difference from normoxia (linear mixed model ANOVA; * indicates $p<0.05$; ** indicates $p<0.01$; *** indicates $p<0.001$).
DOI: https://doi.org/10.7554/eLife.44986.015

• Supplementary file 3. Arterial $P_{O_2}$ and metabolic data collected from the subset of flights during which arterial $P_{O_2}$ was measured (therefore bird 45 only). Values are mean ± SEM. Asterisks indicate significant difference from normoxia (ANOVA; * indicates $p<0.05$; ** indicates $p<0.01$; *** indicates $p<0.001$).
DOI: https://doi.org/10.7554/eLife.44986.016

• Supplementary file 4. Kinematic variables reported as mean ± SEM for four bar-headed geese flown from 12.5 ms$^{-1}$ to 15.0 ms$^{-1}$ in normoxia ($F_iO_2$ = 0.21) and moderate hypoxia ($F_iO_2$ = 0.105). Data in this table are from a separate unpublished study on the same birds as the primary study. Each mean is the average of 40 wingbeat values. Asterisks (*) indicate statistically significant comparisons between normoxia and moderate hypoxia in a mixed model analysis after posthoc application of the false discovery rate. Frequency is calculated as the inverse of the entire wingbeat duration, not in terms of the upstroke or downstroke alone.
DOI: https://doi.org/10.7554/eLife.44986.017

• Transparent reporting form
DOI: https://doi.org/10.7554/eLife.44986.018

## Data availability

All data generated or analyzed during this study are included in the manuscript and supporting files. R script and data files were deposited in Dryad (https://doi.org/10.5061/dryad.fg80hp6).

The following dataset was generated:

| Author(s) | Year | Dataset title | Dataset URL | Database and Identifier |
|---|---|---|---|---|
| Meir JU, York JM, Chua B, Jardine W, Hawkes LA, Milsom WK | 2019 | Data from: Reduced metabolism supports hypoxia flight in the high-flying bar-headed goose (*Anser indicus*) | https://doi.org/10.5061/dryad.fg80hp6 | Dryad Digital Repository, 10.5061/dryad.fg80hp6 |

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
