## [Decision Letter]

Thank you for submitting your article "Reduced metabolism and increased O_2_ pulse support hypoxic flight in the bar-headed goose (*Anser indicus*)" for consideration by *eLife*. Your article has been reviewed by three peer reviewers, and the evaluation has been overseen by a Reviewing Editor and Harry Dietz as the Senior Editor. The following individual involved in review of your submission has agreed to reveal their identity: Jon Harrison (Reviewer #1).

The reviewers have discussed the reviews with one another and the Reviewing Editor has drafted this decision to help you prepare a revised submission.

This paper presents a novel and challenging experimental analysis of the physiology of bar-headed geese flying in anoxic conditions. By training instrumented geese to fly in a wind tunnel (no small feat) this work gives new insight into how these geese make their remarkable flight over the Himalayas. The reviewing editor and the reviewers enjoyed reading this manuscript and believe that the major issues highlighted below can be.

Essential revisions:

The major issues raised in the reviews that must be addressed are:

1) Clarification that there is no evidence that oxygen delivered per heart beat (oxygen pulse) increases if one compares steady-state flight during hypoxia to normoxia within bar-headed geese. This point was raised by two reviewers.

2) Making the data available and addressing the statistical (survivor bias) concerns.

3) There should be discussion of the issue of the minimum cost of flight and the possibility that hypoxic birds "cheated" to remain aloft, since the major finding was that metabolic rate decreased during forward flight in hypoxia.

4) There should be discussion of the possibility of anaerobic metabolism occurring in the hypoxic experiments, which cannot be discounted with the data provided to date.

5) The Abstract is lacking in biological context and too oriented towards a specialist reader. By contrast the cover letter explains clearly the (amazing) natural history in which this work is set and why this is important. It would benefit the lay reader to have this, and the reason for this study, explained in the Abstract. In addition (optionally) it may benefit the authors to modify the title of the paper to give the context in which hypoxic flight occurs (i.e. why is this relevant to this goose?).

Major points:

While a previous paper measured metabolic and heart rates during flight at near sea-level conditions, this is the first to examine these aspects of flight physiology under hypoxic conditions similar to those experienced during their high-altitude migrations. There are several striking and interesting results. The first is that hypoxia significantly reduced flight metabolic rate. This is surprising because it is generally thought that energy use by the flight muscles predominates during flight. A second important finding was that heart rate showed little change during hypoxia. Finally, venous temperature decreased, potentially increasing oxygen loading at the lung. The latter is a novel finding, to my knowledge. The methods are clearly described and appear to be well-done. The only exception is that the method for producing hypoxia (mixing N_2_ and air in the mask) was unable to create a stable background, so V_O2_ was not measured in hypoxia. Another weakness is that the sample sizes were quite low, especially in hypoxia, where it was difficult to get the birds to fly. Nonetheless, I think that the results are generally clear and self-congruent, convincing me that most of the conclusions are sound. However, the final sentence of the Abstract is confusing and may be wrong. It implies that flight in hypoxia is facilitated by an increase in the oxygen pulse. But this parameter actually decreases in hypoxic flight relative to normoxic flight (Table 1, bottom line). And though venous P_O2_ decreases in hypoxic flight, so does arterial P_O2_ (actually more so, at least for the one reported bird, subsection “Blood gases”, third paragraph), fitting with the decreased oxygen pulse in hypoxic flight vs. normoxic flight. Comparing hypoxic steady state flight to normoxic, it seems that the decrease in metabolic rate was similar to the decrease in oxygen pulse. This suggests to me that the only demonstrated mechanism for hypoxic flight is the ability to fly at lower metabolic rate. I do wonder if mechanical work per wing stroke declined, given that wing beat frequency was constant.

One concern of mine that appeared repeatedly is a sort of survivor bias in the presentation of the results, where summary data from the moderate and severe hypoxia treatments are shown together. This is troubling because we know that 3 of the birds in the moderate hypoxia group were unwilling/unable to fly in severe hypoxia and are therefore not directly comparable to the birds that were willing and able. Based on your description in the subsection “Calculations and statistical analysis”, your statistical tests may avoid this problem, but also might not because the birds that did not fly in severe hypoxia are not just "missing data", they were unable/unwilling to fly and that information may not be effectively incorporated in the tests. I would have been happy to investigate further by looking at the source data, but the Dryad data package (listed as doi:10.5061/dryad.fg80hp6) appears to not be available at this time. Figures and tables where presentation appears to be affected by a survivor bias: Figure 2, Figure 3A, Figure 4, Table 1, Supplementary file 2. I suggest you A) discuss this possible problem in the text and B) add some supplementary figures and tables that show only the directly comparable data.

It appears from your results (Figure 2) that the minimum cost of flight is similar for all three oxygen levels. Is this statistically supported? If so, I think it helps clarify your explanation in the last paragraph of the subsection “Effects of Hypoxia” – about how the birds cope with the metabolic challenge – they "become more efficient" because they are restricted to fly in only the most efficient manner; one they occasionally used in normoxic conditions during the experiment and probably the way they'd fly during an actual migration. Alternatively, they may increase efficiency by "cheating" and taking advantage of turbulence or lower-speed regions created by the operator and experimental apparatus, a possibility you should also note.

Whale (2012) is cited several times but as far as I can tell is not publicly available. Please reproduce any relevant figures or tables from this paper in the supplement of this manuscript. If it is publicly available, please include a URL in the reference.

The technical work to produce this study is admirable and must have been exceptionally challenging. This is further evidenced by the low success rate in flying instrumented birds under hypoxic conditions. This is also the study's biggest weakness. Important conclusions are based on the data obtained from only a small few (e.g. moderate hypoxia) or one (severe hypoxia) individuals that consistently performed. Further, methodological problems having to do with the inconsistent mixing of nitrogen and air in the mask mean it is difficult to accept the assumption that V_O2_ data can simply be calculated from V_CO2_ data (i.e. the assumption that RER remained 1.0 across flights and treatments is difficult to accept).

The authors state that they flowed nitrogen directly into the mask at a rate that brought O_2_ levels approximately to *F_i_O_2_* of 0.105 and 0.07. This permits mixing of ambient air and nitrogen in the mask… which is likely a very unstable mixing environment (and may lead to the inability to obtain 'reliable, stable' baseline O_2_ levels in the mask). Why did the authors not mix nitrogen with ambient air upstream of delivery to the mask? Surely this would have resulted in a more consistent and stable *F_i_O_2_*.

The authors dismiss the potential criticism that I allude to below regarding the possibility of tapping into anaerobic metabolism to support flight under hypoxia (or of an induced metabolic acidosis) concluding that there was no apparent oxygen debt to be repaid after the flights had ended. But, they did not derive V_O2_ data from hypoxic flights (and do not present such data for the recovery period). So, this seems to remain a distinct problem.

It is unclear if individual birds had both venous *and* arterial P_O2_ sensors implanted. In the end, only one bird's arterial P_O2_ was successfully recorded. It is thus difficult to judge how representative these data are of other individuals.

---

## [Author Response]

Essential revisions:The major issues raised in the reviews that must be addressed are:1) Clarification that there is no evidence that oxygen delivered per heart beat (oxygen pulse) increases if one compares steady-state flight during hypoxia to normoxia within bar-headed geese. This point was raised by two reviewers.

We have reworded this point throughout the manuscript as “maintaining the increase in O_2_ pulse also measured in normoxia”, which more accurately reflects the data. We apologize that this point was not clear in the original submission.

2) Making the data available and addressing the statistical (survivor bias) concerns.

Apologies for the lack in availability of the data. We misunderstood the Data Dryad system and neglected to provide the temporary DOI link rather than the DOI itself (the data were available in Dryad at the time of submission).

We have added to the Results a discussion of survivor bias which we agree likely bias the severe hypoxia results, but do not bias the normoxia/moderate hypoxia comparisons as the one bird for which we don’t have moderate hypoxic data does not bias the normoxia data.

3) There should be discussion of the issue of the minimum cost of flight and the possibility that hypoxic birds "cheated" to remain aloft, since the major finding was that metabolic rate decreased during forward flight in hypoxia.

We have added some discussion of the possibility that the birds are simply employing the most efficient (minimal cost) flight pattern as suggested by the reviewers (subsection “Effects of Hypoxia”, last paragraph).

4) There should be discussion of the possibility of anaerobic metabolism occurring in the hypoxic experiments, which cannot be discounted with the data provided to date.

This discussion was included in the original manuscript, but has been further edited to clarify (subsection “Effects of Hypoxia”, last paragraph).

5) The Abstract is lacking in biological context and too oriented towards a specialist reader. By contrast the cover letter explains clearly the (amazing) natural history in which this work is set and why this is important. It would benefit the lay reader to have this, and the reason for this study, explained in the Abstract. In addition (optionally) it may benefit the authors to modify the title of the paper to give the context in which hypoxic flight occurs (i.e. why is this relevant to this goose?).

We have significantly modified the Abstract as suggested, and also modified the title.

Major points:While a previous paper measured metabolic and heart rates during flight at near sea-level conditions, this is the first to examine these aspects of flight physiology under hypoxic conditions similar to those experienced during their high-altitude migrations. There are several striking and interesting results. The first is that hypoxia significantly reduced flight metabolic rate. This is surprising because it is generally thought that energy use by the flight muscles predominates during flight. A second important finding was that heart rate showed little change during hypoxia. Finally, venous temperature decreased, potentially increasing oxygen loading at the lung. The latter is a novel finding, to my knowledge. The methods are clearly described and appear to be well-done. The only exception is that the method for producing hypoxia (mixing N_2_ and air in the mask) was unable to create a stable background, so V_O2_ was not measured in hypoxia. Another weakness is that the sample sizes were quite low, especially in hypoxia, where it was difficult to get the birds to fly. Nonetheless, I think that the results are generally clear and self-congruent, convincing me that most of the conclusions are sound. However, the final sentence of the Abstract is confusing and may be wrong. It implies that flight in hypoxia is facilitated by an increase in the oxygen pulse. But this parameter actually decreases in hypoxic flight relative to normoxic flight (Table 1, bottom line). And though venous P_O2_ decreases in hypoxic flight, so does arterial P_O2_ (actually more so, at least for the one reported bird, subsection “Blood gases”, third paragraph), fitting with the decreased oxygen pulse in hypoxic flight vs. normoxic flight. Comparing hypoxic steady state flight to normoxic, it seems that the decrease in metabolic rate was similar to the decrease in oxygen pulse. This suggests to me that the only demonstrated mechanism for hypoxic flight is the ability to fly at lower metabolic rate. I do wonder if mechanical work per wing stroke declined, given that wing beat frequency was constant.

Thank you for these comments. We have reworded the Discussion in regard to oxygen pulse in normoxia vs. hypoxia in order to clarify (see Essential revisions point 1). We have also attempted to emphasize the strengths pointed out here.

One concern of mine that appeared repeatedly is a sort of survivor bias in the presentation of the results, where summary data from the moderate and severe hypoxia treatments are shown together. This is troubling because we know that 3 of the birds in the moderate hypoxia group were unwilling/unable to fly in severe hypoxia and are therefore not directly comparable to the birds that were willing and able. Based on your description in the subsection “Calculations and statistical analysis”, your statistical tests may avoid this problem, but also might not because the birds that did not fly in severe hypoxia are not just "missing data", they were unable/unwilling to fly and that information may not be effectively incorporated in the tests. I would have been happy to investigate further by looking at the source data, but the Dryad data package (listed as doi:10.5061/dryad.fg80hp6) appears to not be available at this time. Figures and tables where presentation appears to be affected by a survivor bias: Figure 2, Figure 3A, Figure 4, Table 1, Supplementary file 2. I suggest you A) discuss this possible problem in the text and B) add some supplementary figures and tables that show only the directly comparable data.

We have added discussion of this issue to the Results, as well as to the figure legends for the figures indicated. We intended Supplementary file 1 to show directly comparable data, as is Figure 2—figure supplement 1 where data are plotted by individual bird for direct comparison. We once again apologize for the lack of data (see comment in “Essential revisions”).

It appears from your results (Figure 2) that the minimum cost of flight is similar for all three oxygen levels. Is this statistically supported? If so, I think it helps clarify your explanation in the last paragraph of the subsection “Effects of Hypoxia” – about how the birds cope with the metabolic challenge – they "become more efficient" because they are restricted to fly in only the most efficient manner; one they occasionally used in normoxic conditions during the experiment and probably the way they'd fly during an actual migration. Alternatively, they may increase efficiency by "cheating" and taking advantage of turbulence or lower-speed regions created by the operator and experimental apparatus, a possibility you should also note.Whale (2012) is cited several times but as far as I can tell is not publicly available. Please reproduce any relevant figures or tables from this paper in the supplement of this manuscript. If it is publicly available, please include a URL in the reference.

Thank you for this insightful observation. We have added a discussion of this possibility to the text, and have added Supplementary file 4 (which contains the flight kinematic data from the Whale reference).

The technical work to produce this study is admirable and must have been exceptionally challenging. This is further evidenced by the low success rate in flying instrumented birds under hypoxic conditions. This is also the study's biggest weakness. Important conclusions are based on the data obtained from only a small few (e.g. moderate hypoxia) or one (severe hypoxia) individuals that consistently performed. Further, methodological problems having to do with the inconsistent mixing of nitrogen and air in the mask mean it is difficult to accept the assumption that V_O2_ data can simply be calculated from V_CO2_ data (i.e. the assumption that RER remained 1.0 across flights and treatments is difficult to accept).

We agree that our dataset would be significantly improved if stable V_O2_ records were obtained in hypoxia. Unfortunately, the methodological limitations of the study prevented this, as was also the case in other published studies (see subsection “Methodological considerations”, last paragraph). We were very straightforward in elucidating these limitations and fully describing the inferences we made throughout the study. We assume that RER remains close to 1.0 in hypoxia (as measured in normoxia) since durations between flights in normoxia and moderate hypoxia were not significantly different (we also added this point to this section in the manuscript). We do expect that RER would fall in longer flights (see the aforementioned paragraph).

The authors state that they flowed nitrogen directly into the mask at a rate that brought O_2_ levels approximately to F_i_O_2_ of 0.105 and 0.07. This permits mixing of ambient air and nitrogen in the mask…which is likely a very unstable mixing environment (and may lead to the inability to obtain 'reliable, stable' baseline O_2_ levels in the mask). Why did the authors not mix nitrogen with ambient air upstream of delivery to the mask? Surely this would have resulted in a more consistent and stable F_i_O_2_.

We were unable to do this because of the method in which we introduced ambient air into the mask (ambient air from the wind tunnel was pulled into the mask and over the nares via space at the top of the mask) – subsection “Physiological measurements”, first paragraph. Thus, there was no way to mix the nitrogen with the ambient air prior to entering the max. During testing to calibrate the hypoxic levels (using a Plaster of Paris goose head mold in the mask), we obtained stable O_2_ levels for both levels of hypoxia using this method, so we believed it would be adequate and result in a stable baseline under all conditions. Unfortunately, with the addition of the actual goose in the setup (perhaps due to movement during flight, see Discussion, third paragraph), we were unable to obtain reliable measures of V_O2_ during flight in hypoxia.

The authors dismiss the potential criticism that I allude to below regarding the possibility of tapping into anaerobic metabolism to support flight under hypoxia (or of an induced metabolic acidosis) concluding that there was no apparent oxygen debt to be repaid after the flights had ended. But, they did not derived V_O2_ data from hypoxic flights (and do not present such data for the recovery period). So, this seems to remain a distinct problem.

We have clarified the language regarding anaerobic metabolism (see previous comment). Also, Supplementary files 2 and 3 include V_O2_ recovery data from the flights in which the birds were instrumented with P_O2_ electrodes.

*It is unclear if individual birds had both venous* and *arterial P_O2_ sensors implanted. In the end, only one bird's arterial P_O2_ was successfully recorded. It is thus difficult to judge how representative these data are of other individuals.*

We have clarified the language here, but it was clearly stated in the Materials and methods that “only one site, either arterial or venous, was targeted per surgery”.